# Apparent nosocomial adaptation of *Enterococcus faecalis* predates the modern hospital era

Anna K. Pöntinen [1✉], Janetta Top[2], Sergio Arredondo-Alonso[1,2], Gerry Tonkin-Hill[3], Ana R. Freitas [4], Carla Novais[4], Rebecca A. Gladstone[1], Maiju Pesonen [5], Rodrigo Meneses [2], Henri Pesonen[1], John A. Lees [6], Dorota Jamrozy [3], Stephen D. Bentley [3], Val F. Lanza[7], Carmen Torres [8], Luisa Peixe[4], Teresa M. Coque[9,10], Julian Parkhill [11,12], Anita C. Schürch [2,14], Rob J. L. Willems[2,14] & Jukka Corander [1,3,13,14✉]

*Enterococcus faecalis* is a commensal and nosocomial pathogen, which is also ubiquitous in animals and insects, representing a classical generalist microorganism. Here, we study *E. faecalis* isolates ranging from the pre-antibiotic era in 1936 up to 2018, covering a large set of host species including wild birds, mammals, healthy humans, and hospitalised patients. We sequence the bacterial genomes using short- and long-read techniques, and identify multiple extant hospital-associated lineages, with last common ancestors dating back as far as the 19th century. We find a population cohesively connected through homologous recombination, a metabolic flexibility despite a small genome size, and a stable large core genome. Our findings indicate that the apparent hospital adaptations found in hospital-associated *E. faecalis* lineages likely predate the "modern hospital" era, suggesting selection in another niche, and underlining the generalist nature of this nosocomial pathogen.

[1] Department of Biostatistics, Faculty of Medicine, University of Oslo, Oslo, Norway. [2] Department of Medical Microbiology, University Medical Center Utrecht, Utrecht, The Netherlands. [3] Parasites and Microbes, Wellcome Sanger Institute, Cambridge, UK. [4] UCIBIO/REQUIMTE, Laboratory of Microbiology, Biological Sciences Department, Faculty of Pharmacy, University of Porto, Porto, Portugal. [5] Oslo Centre for Biostatistics and Epidemiology (OCBE), Oslo University Hospital Research Support Services, Oslo, Norway. [6] MRC Centre for Global Infectious Disease Analysis, Department of Infectious Disease Epidemiology, Imperial College London, London, UK. [7] Bioinformatics Unit, IRYCIS, Madrid, Spain. [8] Department of Food and Agriculture, Area of Biochemistry and Molecular Biology, University of La Rioja, Logroño, Spain. [9] Department of Microbiology, Ramón y Cajal Institute for Health Research Ramón y Cajal University Hospital, Madrid, Spain. [10] CIBER in Epidemiology and Public Health (CIBERESP), Madrid, Spain. [11] Wellcome Sanger Institute, Cambridge, UK. [12] Department of Veterinary Medicine, University of Cambridge, Cambridge, UK. [13] Helsinki Institute of Information Technology, Department of Mathematics and Statistics, University of Helsinki, Helsinki, Finland. [14] These authors contributed equally: Anita C. Schürch, Rob J. L. Willems, Jukka Corander. ✉email: a.k.pontinen@medisin.uio.no; jukka.corander@medisin.uio.no

The divergent patterns of the evolution and ecology of generalist and specialist microbes is a topic of long-standing fascination, dating back to at least the 1980s[1]. In the most basic definition, generalist organisms can exploit multiple host taxa and habitat types, while a specialist is limited to only one or few. Although utilising different habitats by generalists may seem favourable, the ability to maintain homoeostasis across diverse environments will increase costs. The level of plasticity in particular traits—or in their ability to efficiently gain or lose particular traits—will determine whether generalists are able to persist in different habitats, or whether exploring diverse environments of a species may promote the evolution of specialists. The study of bacterial pathogens from this particular perspective has attracted a lot of interest, and the molecular drivers of the traits associated with either lifestyle or jumps between habitats or host types have been under scrutiny for many species of pathogenic bacteria[2], such as *Staphylococcus aureus*[3,4], *Campylobacter coli*[5], *Campylobacter jejuni*[6], *Enterococcus faecium*[7], *Salmonella enterica*[8,9], *Yersinia enterocolitica*[10] and *Yersinia pestis*[11], to name but a few. An earlier phylogenomic study revealed that in contrast to *E. faecium*, little phylogenetic divergence was observed among *Enterococcus faecalis* strains[12], with dispersion of single genotypes over various origins and sampling sites[13], indicative of a generalist lifestyle of this organism. Later, a genome-based characterisation of hospital isolates from the UK and Ireland identified three hospital-associated (HA) lineages L1–L3, within a more diverse population, in which putative virulence and antimicrobial resistance genes (ARGs) were overrepresented[14]. A recent study investigated the ancient patterns of evolution in Enterococci, where one of the conclusions made was that the traits underlying the success of *E. faecium* and *E. faecalis* lineages in modern hospitals were already of selective advantage in their ancestors millions of years ago[15]. It has been concluded in multiple studies that a notable expansion of the genome size by phages, mobile genetic elements (MGEs) and pathogenicity islands (PAIs) is strongly associated with the multi-drug resistant (MDR) and hospital-adapted phenotypes for both of these species (*E. faecium*[7,15]; *E. faecalis*[14–16]). However, previous genomic studies of the generalist *E. faecalis*, which ubiquitously colonises animals and insects[17,18], have been limited in their temporal span and coverage of different host species. Here, armed with a collection of 2027 whole-genome sequences from isolates ranging from the pre-antibiotic era in 1936–2018 and a genetically representative subset of 335 isolates sequenced with long-read technology, we obtain a more nuanced picture of the gain and loss of plasmids in the population and the trends of genome changes that allow this species to adapt to different ecologies including the ecological constraints related to the hospital niche.

## Results

**E. faecalis population is interlinked across different host types.** The genomic data from the *E. faecalis* collection of 2027 isolates, together with a reference mapping-based maximum likelihood (ML) phylogeny, and descriptive metadata, is available as a Microreact project https://microreact.org/project/3T9X5PlUD. The isolate collection represented a wide variety of isolation sources (Fig. 1). The majority were of human origin, with 47.7% (967/2027) from hospitalised patients and 19.3% (391/2027) from non-hospitalised persons. The majority (62.9%, 608/967) of the hospital isolates were retrieved from blood samples, representing true infections instead of random commensal carriage. Sources also included environmental sites (7.7%, 156/2027), farm animals (6.4%, 130/2027) and a subcollection of various other sources (10.8%, 219/2027), such as food products, pet animals and wild mammals. Wild bird isolates comprised 6.7% (136/2027) of the total samples, representing 11 different avian orders. The greatest number of these isolates (35.3%, 48/136) belonged to 15 different species of *Accipitriformes*, many of which species are known to be

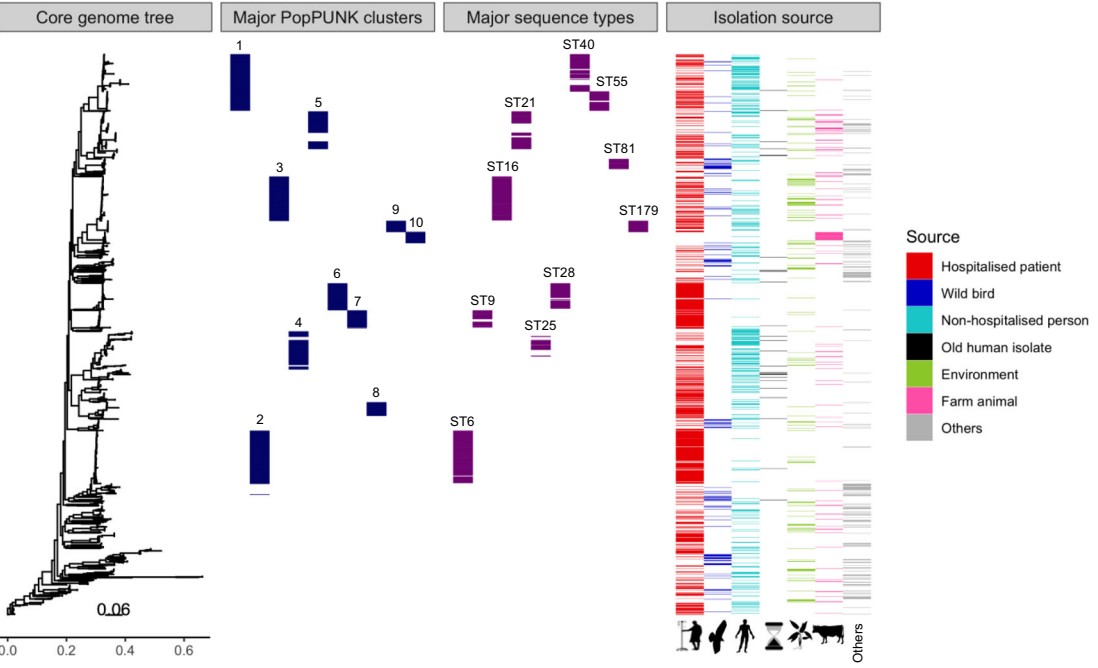

**Fig. 1 Overview of the species-wide *E. faecalis* collection (*n* = 2027) depicts interlinkage of the population across host types.** First panel, maximum likelihood (ML) phylogeny on mapping-based[68] core-genome alignment, estimated by RaxML v.8.2.8[76]. Second and third panels, coloured blocks depicting the ten largest Population Partitioning Using Nucleotide K-mers (PopPUNK)[19] clusters (blue) and multi-locus sequence types (MLSTs) (purple), respectively, against the species-wide phylogeny. Fourth panel, isolates coloured as per their isolation source: hospitalised patient (red), wild bird (dark blue), non-hospitalised person (light blue), old human isolates (black), environment (green), farm animal (pink) and others (grey). Source data are provided as a Source Data file.

migratory or partial migrants. A minority of avian isolates (in total 13.2%, 18/136) were collected from orders *Charadriiformes*, *Anseriformes* and *Columbiformes* that frequently inhabit ecosystems within or in close contact with human settlements. Notably, the collection included 31 *E. faecalis* isolates of human origin collected between 1936 and 1990, the earliest three of which preceded the major commercial antibiotic era in the mid-1940s (Supplementary Fig. 1). Of these, 28 were here defined as old isolates of human origin and three categorised as hospitalised patient, as the information was available for those isolates. Originating in total from 24 countries, the collection covered a global geographic distribution (Supplementary Fig. 1).

To explore the population genomics of *E. faecalis*, the isolates were clustered by using an alignment-free clustering of the core genome using the Population Partitioning Using Nucleotide K-mers (PopPUNK) software[19], which delineated the *E. faecalis* collection into 173 primary clusters. The ten largest clusters comprised more than half of the total isolates (52.7%, 1068/2027) (Fig. 1). Of the total clusters, 57 were identified by PopPUNK as genetically isolated groups, comprised of single isolates. The top three largest PopPUNK clusters (PP) overlapped with three major sequence types (STs) in *E. faecalis*, ST6 (PP2), ST16 (PP3) and ST40 (PP1), of which the two latter have been isolated from various sources both human and non-human, while ST6 has been described as human-associated[20–24]. Phylogenetically closely related to PP3, PP9 exclusively included ST179, a human-associated single-locus variant of ST16[20], and PP10 solely contained ST58 and ST63. The other PP clusters included a broader set of STs, of up to 12 different STs in PP4, with the most prevalent being ST21 and ST22 (PP5), ST28 (PP6), ST9 (PP7) and ST8 and ST64 (PP8). Along with the human-associated ST6, the major types ST9, ST28 and ST64 have been shown to be enriched in clinical isolates[25]. However, species-wide mapping-based phylogenetic construction of the present collection did not identify any deep branches separating large host-specific subpopulations, as would be expected in the case of any high-level host specialisation of particular sublineages[3]. On the contrary, the aligned distribution of isolation sources illustrated the interlinkage of *E. faecalis* population across widely different host types (Fig. 1).

**Pangenomic variation shows no signal of host specialisation.** In order to shed light on the generalist lifestyle of *E. faecalis*, we investigated the core and accessory genome contents across different host types and contrasted this with *E. faecium*. Pangenomes for 2026 *E. faecalis* and 1602 *E. faecium*[7] isolates that passed the assembly and annotation pipelines were defined using Panaroo[26]. Pangenome estimation revealed a comparatively large core within a moderate-sized genome for *E. faecalis*, and Heap's law alpha value of 2.16 suggested a closed pangenome[27]. Of the 15,827 total genes, 13.1% (2068) were core genes, amounting to 76.1% (2068/2717) of the average genome size. The *E. faecalis* pangenome was of similar size to that of *E. faecium*, which totalled 16,711 genes. However, *E. faecalis* presented a notably larger core genome, as *E. faecium* harboured only 10.0% (1665) core genes of the total, representing 62.3% (1665/2674) of its average genome size.

To encapsulate genomic information elucidating evolutionary processes towards the generalist lifestyle, potentially neglected by core-genome variation only, the entire *E. faecalis* collection was additionally clustered by using Pangenome Neighbour Identification for Bacterial Populations (PANINI)[28]. PANINI analysis showed that the clusters defined by accessory gene content largely coincided with PopPUNK clusters, which had been defined by using the core-genome-only option (Fig. 2a).

Aligning the PANINI cluster network with the two major isolation sources illustrated how the *E. faecalis* isolates of hospital patient and avian origin were not diverged into separate clades but largely embedded within the same clusters (Fig. 2b, c), indicating no prominent host specialisation. Of note, the old *E. faecalis* human isolates were distributed across extant clusters (Fig. 2d). Although a few accessory genome clusters mainly consisted of hospital isolates, network analyses of the accessory gene contents did not identify any comprehensive host-associated clustering for *E. faecalis* (Fig. 3), consistent with both PANINI and PopPUNK clustering. A few small clusters of environment, wild bird, and farm animal origin presented host specificity that was identified to derive from sampling of the isolates from the same location and time period. However, *E. faecium* presented multiple clusters, based on the accessory gene contents, clearly associated with hospital origin (Supplementary Fig. 2), correlating with the previously described HA structure within the *E. faecium* population[7,29]. Further supporting the idea of the generalist nature of *E. faecalis*, accessory gene frequencies presented similar distributions across all major host types. There were no significant differences (one-sided permutation tests, $P > 0.20$) between empirical cumulative distribution function (CDFs) of HA isolates when compared to other major host types within the *E. faecalis* population (Supplementary Fig. 3). In contrast, the empirical CDFs for accessory gene frequency distributions of HA *E. faecium* isolates presented a significant difference (one-sided permutation tests, $P < 0.05$) to those of non-hospital origin (Supplementary Fig. 4).

**Population chromosome and plasmidome reveal stable genome sizes and historical resistance traits.** Genome size predictions based on 335 *E. faecalis* genomes with fully contiguous assemblies showed an average genome size of 2.97 Mbp for the chromosome and 24.3 kbp for the plasmidome. Chromosome comparison of the hybrid assemblies showed largely invariable sizes across the major host types (Supplementary Fig. 5). While non-hospital origin and those defined as others presented smaller mean sizes (one-way ANOVA, $P < 0.001$), as compared to that of hospital origin, generally the chromosome sizes did not differ even between widely different host types, such as hospitalised patients and wild birds, reflecting the above-described generalist lifestyle. Temporal analysis revealed that neither the overall chromosome nor the population plasmidome size increased over time (Fig. 4a), not even in the hospital environment (Supplementary Fig. 6), unlike what has been observed for other nosocomial pathogens such as *E. faecium*[7,15,30]. However, when the data are stratified according to the source and in particular HA clusters, a weak intermittently increasing trend of predicted plasmid content size could be detected in HA clusters (Supplementary Fig. 6). The agreement of both hybrid assemblies and predictions returning comparable chromosome and plasmidome sizes over years also illustrated successful performance by mlplasmids (Fig. 4a). In agreement with the depicted generalism of *E. faecalis*, comparisons of the distributions of shared sequence content (calculated using mash) based on the plasmidome and chromosome predictions showed no signs of host specialisation, as the different host types were seen blended in the hierarchical clustering dendrograms (Fig. 4b). Although plasmids are abundant in Enterococci, with the hybrid assemblies and predictions combined, we found a high proportion (27.7%, 562/2027) across the entire collection of the *E. faecalis* isolates to lack plasmid sequences. Notably, these isolates covered the whole isolation year range from 1936 to 2018 and all the major isolation sources, including isolates of both hospital and non-hospital origin, with similar proportions of 23.3% (225/967) and 25.8% (101/391),

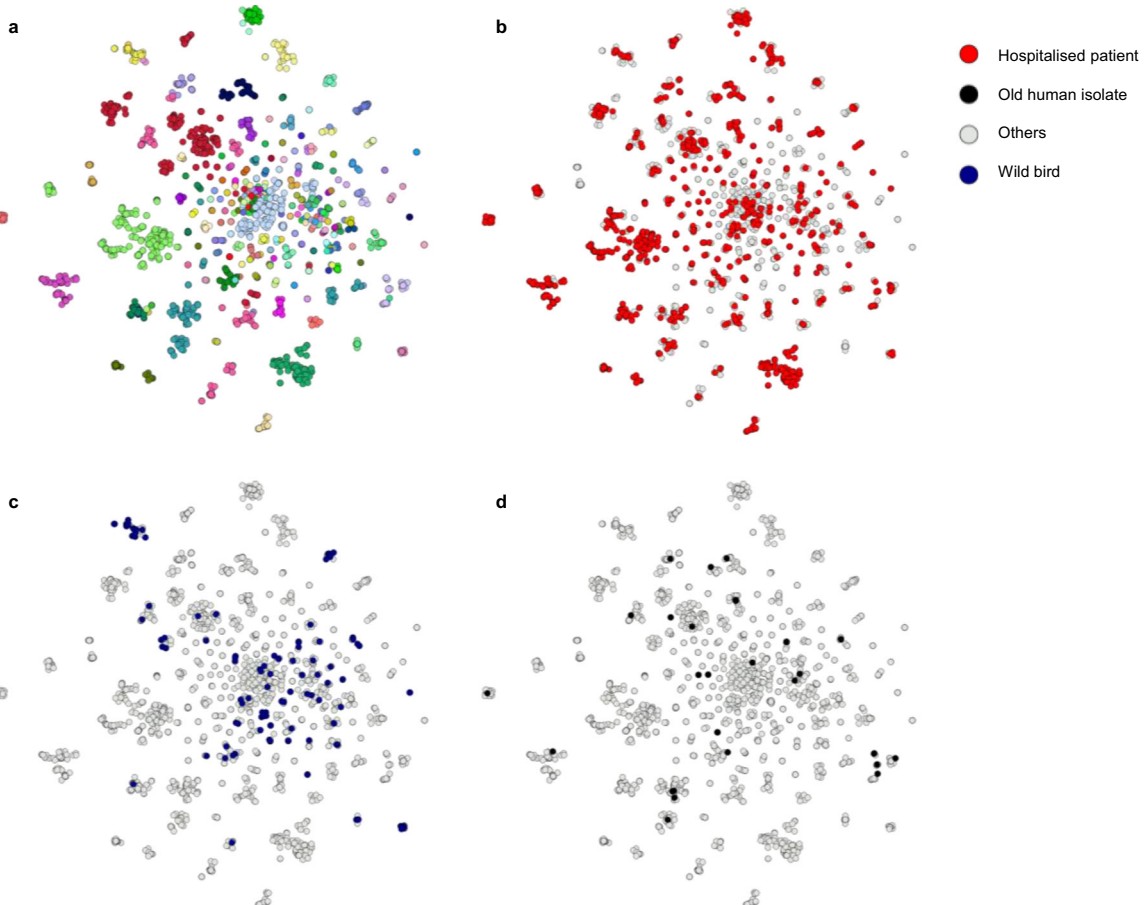

**Fig. 2 Pangenome Neighbour Identification for Bacterial Populations (PANINI)[28] networks on *E. faecalis* pangenome depict isolates of different origins distributed across the clade structure and embedded within the same, clearly defined extant clusters. a** PANINI network depicts accessory clusters predominantly following the designated Population Partitioning Using Nucleotide K-mers (PopPUNK; PP)[19] clustering, with each node representing a single isolate (*n* = 2027), each group of attached nodes largely representing a PANINI cluster (*n* = 70), and each colour indicating a separate PP cluster (*n* = 173). **b** Hospitalised patient isolates shown in red and other isolation sources in grey. **c** Wild bird isolates shown in dark blue and other isolation sources in grey. **d** Old isolates of human origin shown in black and other isolation sources in grey. The illustration was created using Microreact[77]. Source data are provided as a Source Data file.

respectively. As compared to the previously described 35% of avian *E. faecalis* isolates[20], here even a higher proportion (41.9%; 57/136) of isolates lacking plasmids was found among wild bird isolates.

To increase the temporal resolution of plasmidome evolutionary trends over a time span of 1940–1985, 30 old isolates were included in the long-read Oxford Nanopore Technologies (ONT) sequencing scheme. Of these, seven isolates lacking plasmids were excluded from further analysis. Alignment-free k-mer-based clustering of the remaining 23 isolates by using Mashtree[31] revealed a diverse genomic background, except for two indistinguishable pairs (Supplementary Fig. 7). In total, 45 plasmids were identified, one to four per isolate and varying in size from 2.2 to 123 kbp. K-mer presence/absence clustering of these plasmids revealed a large diversity (Supplementary Fig. 8; https://microreact. org/project/110-nP5qs). No clustering based on isolation years or presence of virulence genes was observed, although we identified putative virulence genes in 22/45 plasmids, earliest of which was from 1946. These included the aggregation substance *asa1* and *asa1-like*, the cytolysin gene cluster and a bile salt hydrolase (*bsh*). Only three plasmids harboured ARGs, including E07284_3 (1960) encoding tetracycline resistance (*tet(L)*), E07292_2 (1962) encoding chloramphenicol resistance (*cat*), and E00740_3 (1985) encoding erythromycin (*erm(B)*), trimethoprim (*drfC*), bleomycin (*ble*) and

two aminoglycoside resistance genes (*aac(6′)-aph(2″)* and *aadD*) (Fig. 4c). These depicted notably early emergence of antimicrobial resistance (AMR)-conferring traits in the *E. faecalis* population. In addition, plasmid E07292_2 (80 kbp, The Netherlands, 1962) also encoded genes annotated as bacteriocin, relA/B toxin/antitoxin system, and two gene clusters encoding for arsenical and mercury resistance. This region of ~20 kbp is identical to parts of plasmids from strains ATCC29212 (the UK, 1904, "CP008814 [https://www. ncbi.nlm.nih.gov/nuccore/CP008814.1/]") and T2 (Japan, before 1992, "NZ_GG692853 [https://www.ncbi.nlm.nih.gov/nuccore/ NZ_GG692853]") (Supplementary Fig. 9). The isolate from 1904 contained no antibiotic resistance genes. Similarity analysis of the E07292_2 *cat* gene revealed 100% identity to *cat* from other *E. faecalis* strains and also *E. faecium* and different species of Staphylococci, suggesting interspecies exchange of genes.

The presence of the gene clusters encoding for arsenical and mercury tolerance or resistance is of interest as they appear not only in low prevalence (6.3%, 128/2027) in the current strain collection, but are also restricted to mainly three countries, Portugal (35.2%, 45/128), Spain (31.3%, 40/128) and The Netherlands (22.7%, 29/128), and to only 13.3% (23/173) of PP clusters. PP clusters containing more than ten isolates harbouring these gene clusters included PP1 (12.5%, 16/128), PP5 and PP16 (both 11.7%, 15/128), and the HA PP clusters PP7 and PP18

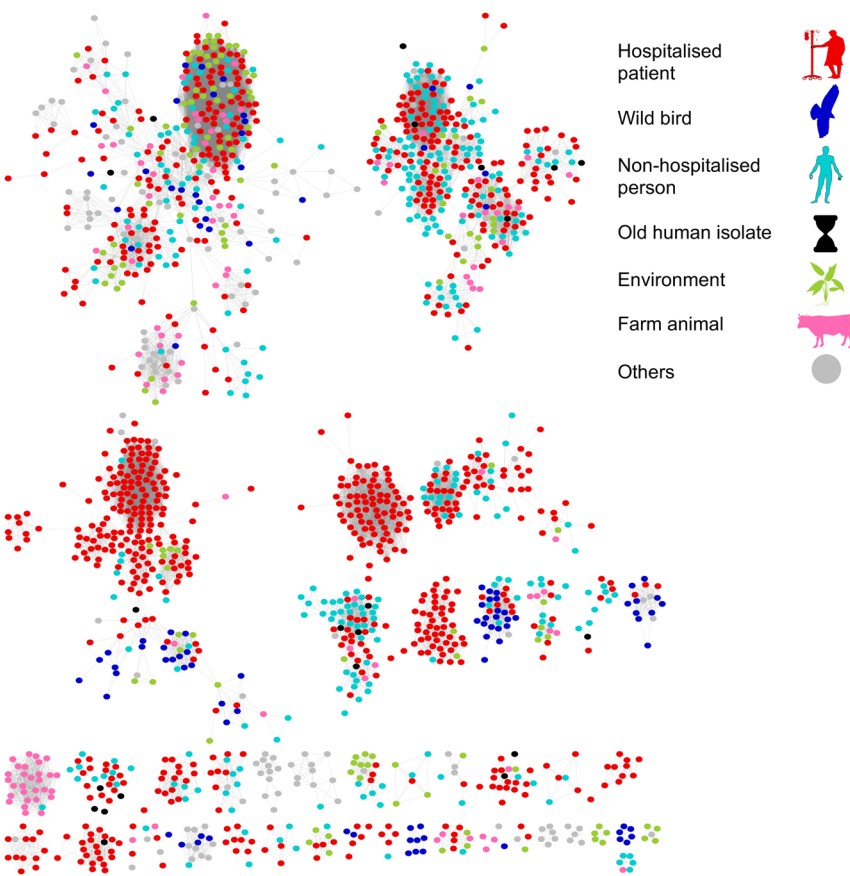

**Fig. 3 Network analysis of the *E. faecalis* accessory genomes, as defined by Panaroo[26], depicts no prominent host-specific clustering.** Nodes indicate separate isolates, connected when shared ≥95% of their accessory genome, and colour-coded according to their isolation sources as indicated in the legend: hospitalised patient (red), wild bird (dark blue), non-hospitalised person (light blue), old human isolates (black), environment (green), farm animal (pink) and others (grey). Components of less than five isolates were filtered out, and the resulting network was visualised using Cytoscape[78]. Source data are provided as a Source Data file.

(21.9%, 28/128 and 9.4%, 12/128, respectively). In addition, we observed co-clustering of isolates from different countries, isolation sources and years within the PP clusters, suggesting that those strains are very stable over time, and that these metal resistance genes are not easily exchanged between different lineages. However, more in-depth investigation of the evolutionary trends of plasmids over the recent years is still warranted.

**Early emergence of hospital-associated lineages.** By molecular dating, we found extant HA clusters dating back to as early as the 19th century and thereby predating the era of modern hospital settings and treatments (Supplementary Table 1). Constant, exponential and Bayesian skyline tree models all gave closely comparable estimates for each cluster. The clusters defined as HA (PP2, PP6, PP7, PP18 and PP20) each included >20 isolates per cluster, of which ≥90% were from hospitalised patients. Sampling bias was estimated negligible given the vast geographic spread of the isolates and the solid quality of temporal signal. Altogether, the hospital isolates in these clusters covered 38.8% (375/967) of the total hospital isolates in the collection. The major STs encompassed in these clusters were ST6 (PP2), ST28 (PP6), ST9 (PP7), ST159 and ST525 (PP18), and ST41 (PP20). Of these, ST6, ST9 and ST28 have been shown clearly enriched in nosocomial infections[14,32], and also isolates of the three latter STs have previously been found in clinical cases[21,33]. Based on molecular dating with constant tree prior, the most recent common ancestor (tMRCA) for the oldest cluster PP2 was estimated to date back to 1846 (95% highest posterior density (HPD) interval 1823–1866;

Fig. 5a) and to 1967 (95% HPD 1961–1973) for the youngest cluster PP6 (Fig. 5b). For PP18, tMRCA was dated to 1917 (95% HPD 1891–1941; Fig. 5c). Temporal estimation on a subcluster of 55 isolates using TempEst[34] timed a tMRCA for PP7 of 1928, while its dating failed in both least-squares and Bayesian molecular dating, most plausibly due to the confounding effect of a large clade with short branches dominating the cluster phylogeny. Of note, of the extant predominant HA clusters, PP6 included amongst others an isolate originating from wild bird (Supplementary Fig. 10). In addition, a number of non-HA clusters, such as PP11 (Supplementary Fig. 10), included multiple isolates of migratory birds embedded among sporadic isolates of hospital origin, again depicting the generalist ability of *E. faecalis* to dwell in largely different host types.

Gene content comparison of HA clusters and commensal isolates identified a total of 84 genes with significantly different frequencies in either of the two categories (Supplementary Data 1). Of these, 78 were enriched in the HA clusters. The majority of hits were linked to MGEs, such as IS elements, integrated plasmids, other plasmid-related elements and phages. Specifically, 23 genes were encoded on a common *E. faecalis* PAI, which carries virulence traits such as surface protein-encoding *esp* and cytolysin-encoding *cyl* and has shown a great capacity of modulating virulence through high frequency of variation[35]. However, none of the enriched genes presented complete presence in HA clusters and absence in others, and, therefore, these genes are likely not to be a strict prerequisite for survival in the hospital settings.

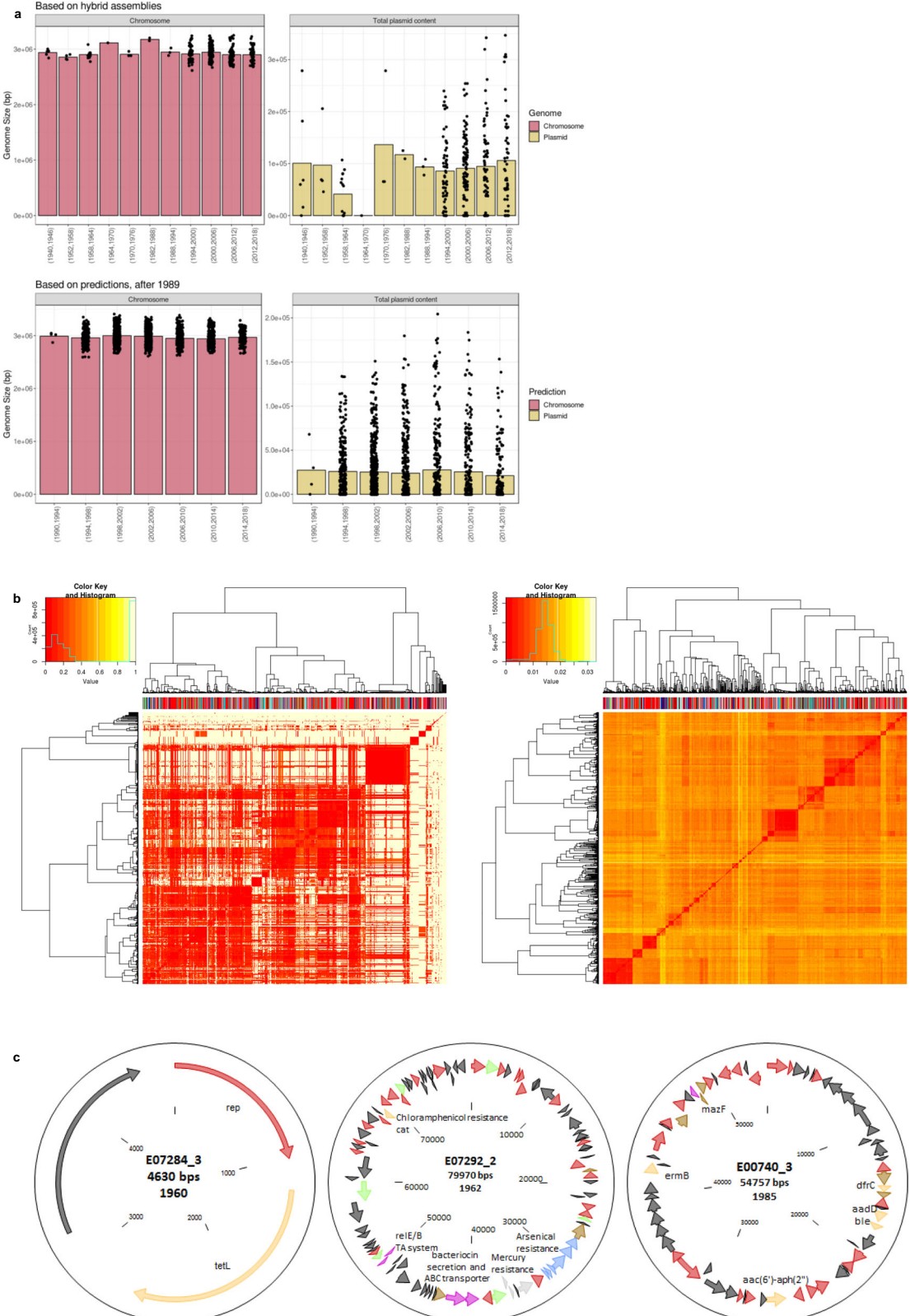

## Evidence of recombination hotspots in hospital-associated lineages

In order to investigate whether the dated HA clusters PP2, PP6, PP7 and PP18 contain particular signatures of recombination, depicting genomic plasticity, we performed recombination analyses using Gubbins[36], with either the old *E. faecalis* isolate E07132 (ST97; Supplementary Fig. 11) or *E. faecalis* V583 (ST6; Supplementary Fig. 12) as a reference. For both,

very similar recombination regions were identified across the PP clusters, although the numbers differed with 8–9 sites in the V583 compared to 2–3 in the E07132 analysis. Closer examination revealed that most of the recombination sites were flanked by the presence/absence of phage-associated gene clusters, IS elements or t-RNA genes. Therefore, we determined all phage(-like) elements using Phaster[37] and plotted them and the recombination

**Fig. 4 Population chromosome and plasmidome show stable genome sizes (bp) over time and plasmid-borne resistance traits in old *E. faecalis* isolates. a** Chromosome (pink) and plasmidome (yellow-green) sizes show no increase over the years of isolation for hybrid assemblies (top panel) or predictions (bottom panel), as derived from mlplasmids[79]. Bar plots represent mean genome sizes (bp), and each black node represents a single isolate. Years are shown in intervals of 6 and 4 years for hybrid assemblies and predictions, respectively. **b** Mash distances ($k = 21$, $s = 1000$) based on the plasmidome (left panel) and chromosome (right panel) predictions show no prominent host specialisation in *E. faecalis*. Isolation source is depicted on top of the dendrogram: hospitalised patient (red), wild bird (dark blue), non-hospitalised person (light blue), old human isolates (black), environment (green), farm animal (pink) and others (grey). Dissimilarity matrices of the isolates are depicted as heat maps, coloured as indicated in the colour key. Histograms on top the colour keys show distributions of Mash distances. Source data are provided as a Source Data file. **c** Oxford Nanopore Technologies (ONT) sequencing of the old *E. faecalis* isolates revealed old plasmids carrying multiple antimicrobial and metal resistance genes. Plasmid lengths and isolation years are indicated under the plasmid name. Red arrows indicate plasmid-associated genes and transposases, yellow arrows indicate antimicrobial resistance genes; light grey, mercury resistance and light blue, arsenical resistance genes.

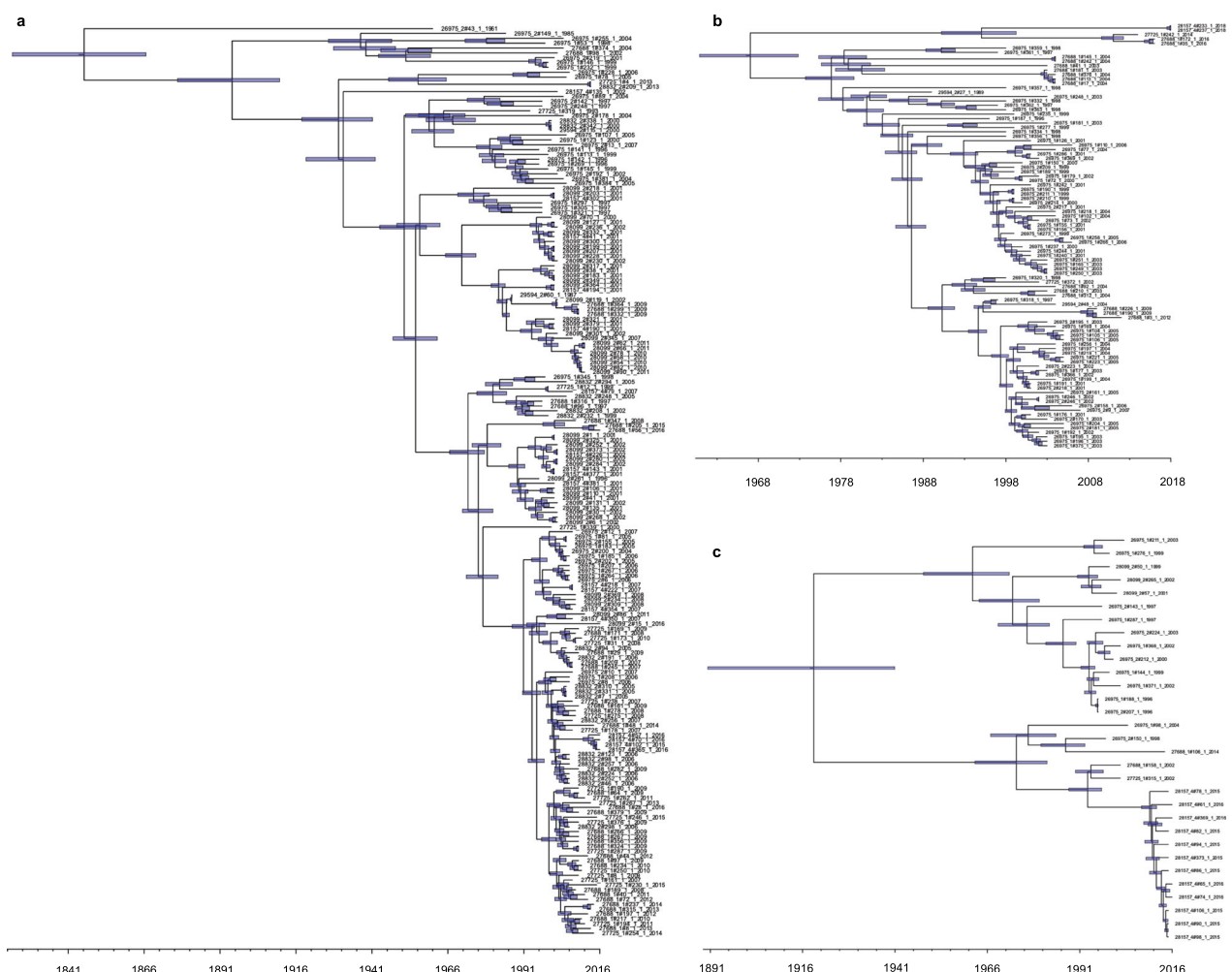

**Fig. 5 Molecular clock dating reveals early and repeated emergence of extant hospital-associated (HA) E. faecalis Population Partitioning Using Nucleotide K-mers (PopPUNK; PP)[19] clusters. a** PP2 ($n = 189$). **b** PP6 ($n = 97$). **c** PP18 ($n = 31$). Dated maximum clade credibility trees were estimated based on median values from Bayesian Evolutionary Analysis by Sampling Trees (BEAST2) v.2.5.0[83-85] using strict clock model and constant tree. Blue error bars represent the 95% highest probability density (HPD) intervals of median tree heights. The time scale is shown in years. Source data are provided as a Source Data file.

sites on the genome maps of both references. In addition, we performed a reciprocal BLAST of E07132 against V583 and mapped those as well. It is clear that all three recombination sites for E07132 and 6/9 for V583 align with phage elements (Supplementary Fig. 13). Analysis of the remaining three sites for V583 revealed that two regions contained a (putatively) integrated plasmid, including one also encoding the virulence-associated aggregation substance. Although this element was absent in the majority of the HA strains, we often observed insertions of other elements at the same position, suggesting a

hotspot of integration. Similarity search also revealed phage-like genes in the last region, suggesting phage integration, unrecognised by Phaster.

We used long-read sequenced strains to examine in more detail the recombination sites in relation to the genomic organisation of strains per HA PP cluster. Interestingly, among some of the isolates in all four clusters, we found genomic rearrangements at the recombination sites, as exemplified in the MAUVE analysis[38] on a selection of PP18 isolates (Supplementary Fig. 14). In addition, in a number of cases the genomic maps of these isolates

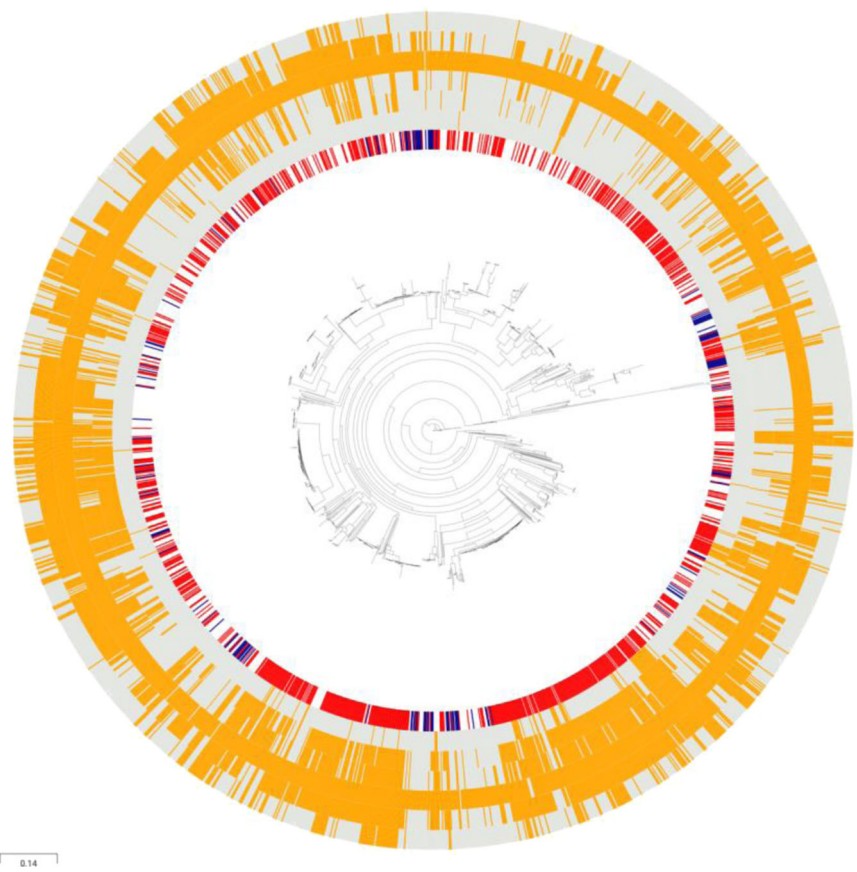

**Fig. 6 Collection-wide antimicrobial resistance gene (ARG) patterns of *E. faecalis* depict abundance of antimicrobial resistance (AMR), also in isolates representing widely different host types.** Presence of acquired and apparent intrinsic ARGs is aligned with the species-wide (*n* = 2027) reference mapping-based maximum likelihood phylogeny. Inner ring, selected isolation source: hospitalised patient (red) and wild bird (dark blue). Outer rings, presence (orange) and absence (grey) of selected major ARG classes as defined by using Antimicrobial Resistance Identification By Assembly (ARIBA) v.2.14.4[88] against ResFinder 3.2 database[81], from inner to outermost ring: vancomycin (glycopeptide), aminoglycosides, macrolides, lincosamides, tetracyclines and phenicols. In addition, five isolates were identified carrying acquired ARGs to linezolid, of which one was *cfrD*, two were *optrA* and two *poxtA*. The illustration was created using Microreact[77]. Source data are provided as a Source Data file.

revealed genomic rearrangement to result in replichore imbalance as exemplified by an isolate from PP7 and PP18 (Supplementary Fig. 15). Although isolated at the same hospital and in the same year, they presented differences in their accessory genomes, including differences in plasmid content, suggesting that the imbalance arose independently in each cluster.

**Early emergence and dissemination of antimicrobial resistance and virulence factors in the *E. faecalis* population.** Collection-wide screening of acquired and apparent intrinsic ARGs illustrated the abundance and wide distribution of AMR traits among *E. faecalis* isolates. For clarity, we depicted the main resultant ARGs as profiles of vancomycin (glycopeptide), aminoglycosides, macrolides, lincosamides, tetracyclines and phenicols, and aligned them with the species-wide phylogeny (Fig. 6). Vancomycin resistance-conferring genes are prevalent in *E. faecium*[39,40] but less frequent in *E. faecalis* and were harboured by 4.9% (100/2027) of the total isolates in the present collection. However, all of the *van*-carrying isolates also harboured resistance to at least two other major antimicrobial classes and 27.0% (27/100) to all five of them, depicting the worryingly broad dissemination of ARGs in *E. faecalis*. Apart from vancomycin, 18.4% (372/2027) of the total isolates harboured genes conferring resistance to all other major classes, while only eight isolates lacked all ARGs screened (Supplementary Data 2). The majority

of the former, 53.8% (200/372), were hospital-derived human isolates. However, presence of ARGs was also frequent in wild bird isolates, with only two of them lacking all ARGs screened, potentially indicating how anthropogenic factors facilitate the dissemination of resistance elements widely across dozens of host species. Although none of the wild bird isolates harboured *van* genes and only 11 strains carried ARGs to phenicols, resistance to other antibiotic classes proved to be abundant among them, as 23.5% (32/136) harboured ARGs to all four other major classes and 39.0% (53/136) to at least three of them. Nearly all isolates of the total (99.3%, 2012/2027) harboured resistance genes to lincosamides, mainly ABC protein homologue Lsa-encoding *lsaA*, which is prevalent in *E. faecalis* and confers intrinsic resistance to clindamycin, dalfopristin and quinupristin-dalfopristin[41]. ARGs to lincosamides were abundant in isolates of not only human origin but also environmental and animal origin, thereby concluding that lincosamide resistance is indeed a species-wide feature in *E. faecalis*[41]. In addition, genes conferring resistance to tetracyclines (69.3%, 1404/2027), macrolides (47.2%, 957/2027), aminoglycosides (45.0%, 912/2027) and phenicols (21.1%, 428/2027) were frequent in the present collection. In total five isolates of environmental, hospitalised patient and unknown origin were found to harbour acquired resistance genes to the last-resort antimicrobial agent linezolid. In addition to acquired and apparent intrinsic genes, to gain information on the resistance to the frontline agent daptomycin, the collection was screened for

the in-frame deletions in *liaF*, *gdp* or *cls*, shown to confer daptomycin resistance in *E. faecalis*[42]. While the present collection proved to be negative for these deletions, resistance to daptomycin is plausibly driven by much more complex changes in the membrane stress response network[43].

Apart from the lincosamide resistance-conferring *lsaA*, present in the oldest isolates, the emergence of ARGs in our collection was seen in 1960 in two isolates of human origin harbouring ARGs to tetracyclines. One of these was also the earliest plasmid-borne ARG in the collection (Fig. 4c). We found early emerging plasmid-borne traits also in an isolate from 1962 carrying chloramphenicol resistance gene (*cat*) and another from the mid-1980s harbouring erythromycin resistance-conferring *erm(B)* (Fig. 4c). Resistance to vancomycin was seen emerging in two isolates of human (*vanA*-type) and bovine (*vanB*-type) origin from 1987, which coincides with the first reported human clinical cases of vancomycin-resistant Enterococci (VRE)[44,45], while predating the first finding of enterococcal vancomycin resistance in an isolate of non-human origin[46]. Notably, both of these isolates resided in the same HA cluster, further indicating the potential for enterococcal AMR traits to circulate across host boundaries.

Collection-wide screening also showed an abundance of virulence genes, regardless of isolation source (Supplementary Fig. 16 and Supplementary Data 2). However, this exploratory analysis did not indicate any systematic difference in known virulence factors between HA clusters and isolates from other sources. Nearly half of the total resultant gene variants (11/23) were found in >90% of the isolates, of which genes *srtA*, *cCF10* and *cOB1*, were found in all. Furthermore, virulence genes were widely seen in the oldest of isolates, and even the latest variant to emerge was seen in an isolate from as early as 1961.

## Discussion

An ecological generalist microbe must carry genes or adaptive variants across lineages that enable it to survive in multiple niches, as opposed to specialist organisms that can partition these adaptive genes into specialist lineages. When investigating a large collection from widely different sources, we found the generalist nature of *E. faecalis* to be clearly supported by pangenomic clustering depicting no prominent host specialisation. Although core-genome clustering revealed some apparent HA clusters, clearly specialist subpopulations were not discovered, as accessory gene content networks showed that various host types were distributed across distinct clusters. On top of hundreds of long-read sequences, successful chromosome and plasmidome predictions provided further evidence on the generalist nature of *E. faecalis*, as stable genome sizes were identified across the wide range of isolation years and sources. *E. faecium*, on the contrary, clearly showed host-specific HA subpopulations, consistent with the previously described host-specific clusters[7,29]. In *E. faecium*, the evolution of HA subpopulations coincided with increased genome sizes in hospital-acquired isolates[7]. Also, recently in *S. aureus*, host-associated accessory gene clusters were discovered across diverse clonal complexes and thus independent of core clustering, suggesting host-specific gene pools for adaptation in this multi-host pathogen[3].

A recent study of the evolutionary history of Enterococci pointed out that the features, such as efficient colonisation and resistance to environmental stressors, which enable Enterococci to prevail as leading nosocomial pathogens in the modern hospital surroundings, may date back to their Palaeozoic ancestors[15]. We thus sought to establish the temporal emergence of the apparent HA clusters in the present *E. faecalis* collection with aid of molecular dating. We found the oldest of the HA clusters

(PP2 that includes ST6 isolates), to date all the way back to the mid-19th century, markedly predating the emergence of modern hospital settings. Furthermore, the oldest clusters were shown to predate the introduction of antibiotics as a common treatment, while the most recent HA cluster (PP6), emerged only a few decades ago. Thus clustering and dating analyses indicated that apparent adaptation to the HA niche is actually likely to be due to selection for survival in a broader set of niches, consistent with *E. faecalis* being a generalist, rather than a specialist. In the apparent HA clusters, a number of recombination hotspots were discovered. In *E. faecium*, genomic plasticity and horizontal gene transfer (HGT) were proven to be a significant evolutionary driver of diversity and adaptation, and recombinant core-genome regions have been specifically identified in HA isolates[47].

MGEs have been known for many decades to disseminate AMR and virulence traits across interspecies barriers among pathogenic bacteria via HGT[7,48,49]. Enterococci, in particular, act as hubs for MGEs, conveying AMR traits among both gram-positive and -negative species[50], most notably *S. aureus*[51] and Streptococci[52,53]. Armed with both short- and long-read sequencing techniques and an isolate collection spanning nearly a century, we were able to identify plasmid-borne AMR traits in *E. faecalis* isolates dating back to the early days of antimicrobial use. While the first reports on AMR predate the commercial use of antibiotics[54], extensive use of antimicrobials has still contributed to the acquired AMR in many clinically relevant pathogens[55] such as methicillin-resistant *S. aureus* (MRSA)[56] and VRE[44]. Considering its generalist nature combined with the fact that *E. faecalis* dwells in surroundings such as human and animal intestines, where it easily comes in contact with a myriad of other species and is readily able to pass on such traits to them, the emergence and increase of novel AMR traits is of substantial importance particularly in Enterococci. Recent findings also suggest that host–plasmid co-evolution, driven by antimicrobial selection, may lead to the emergence and persistence of MDR and even induce MDR after removal of antibiotics[57]. While MDR Enterococci are often overrepresented in HA lineages[14,30], our data also revealed AMR traits to be highly prevalent in *E. faecalis* isolates from non-hospital settings and of non-human origin, again consistent with it being a generalist organism where lineages can survive across multiple niches. Traits conveying AMR have been shown to be abundant in *E. faecalis* isolates from migratory wild birds[20], and we accordingly detected majority of wild bird isolates in the present collection to carry multiple ARGs, unlike what has been found even in domesticated poultry when screening for ARGs in *S. aureus* isolates[3]. Intriguingly, flightless birds in locations remote from anthropogenic impact carried *E. faecalis* isolates lacking other but intrinsic ARGs, in agreement with lack of selection driven by human-applied antimicrobials in the population and its environment[58]. Although selective pressure due to direct antibiotic usage in wild birds may be considered limited, many wild bird species dwelling in human-adjacent locations may be influenced by pharmaceutical pollution and other human activity through shared habitats. However, as we found the ARG-carrying wild bird isolates to be interlinked with HA isolates, it may suggest flying migratory birds act as liaisons between widely different hosts, potentially serving both as reservoirs and vectors of AMR *E. faecalis*.

Besides ARGs, we identified mercury and arsenical resistance genes encoded on MGEs in old isolates, but to a limited extent in more recent isolates, considering multiple countries and isolate clusters. Both metals are naturally present in the environment[59] and were also extensively used as topical disinfectant and antiseptic in hospitals and in the community after early 1900s[60]. Despite that we did not observe a widespread dissemination of these gene clusters in more recent samples, it is possible that they

still circulate in the population at low frequencies, while the extent of ongoing selection pressure maintaining them is unknown. Comparable to AMR traits, it is also very likely that *E. faecalis* exchanged these gene clusters with, e.g. *S. aureus* or *Staphylococcus epidermidis*, as we also identified the mercury gene cluster in *S. aureus* strains from before 1968[61,62] and in two *S. epidermidis* isolates from the beginning of the last century ("UHDC01000002" and "SUM14299").

In summary, the powerful combination of short- and long-read sequencing technologies applied to an ecologically, geographically and temporally extensive sample collection has allowed us to paint a detailed portrait of the evolution of a major pathogen population. We anticipate that the approach adopted here may be generally fruitful for unravelling evolutionary trends and their interplay with ecology in bacterial populations.

## Methods

**Isolate collection.** In total, 2027 *E. faecalis* isolates were used in this study, representing collections from University Medical Center Utrecht, Utrecht, The Netherlands (*n* = 535); European Network for Antibiotic Resistance and Epidemiology at the University Medical Center Utrecht, Utrecht, The Netherlands (*n* = 78); Hospital Ramòn y Cajal, Madrid, Spain (*n* = 503); University of Porto, Porto, Portugal (*n* = 671); and University of La Rioja, La Rioja, Spain (*n* = 240). The total study collection was aimed to cover a large set of clinical human samples, with approximately half of the total samples isolated from hospitalised patients (*n* = 967), in addition to non-hospitalised persons (*n* = 391). Specifically, the collection included a subset of old human isolates (*n* = 28), some of which preceded the modern antibiotic era. To additionally cover a wide range of non-human sources, environmental sites (*n* = 156), farm animals (*n* = 130) and a variety of others (*n* = 219), such as food samples and isolates from pet animals and wild mammals, were included. A large set of wild bird isolates (*n* = 136) was deemed an important asset in investigating in more detail the potential carriage and transmission of ARGs within migratory birds. In total, the isolate collection represented a wide geographic and temporal distribution, isolated in 24 countries and years 1936–2018, respectively (Supplementary Fig. 1; https://microreact.org/project/3T9X5PlUD).

**External data.** For comparison of pangenome estimation and pangenomic variation, Illumina reads of 1644 *E. faecium* isolates were retrieved from the European Nucleotide Archive (ENA) ("PRJEB28495")[7].

**Whole-genome sequencing**

*Short-read sequencing.* Genomic DNA sequencing of the complete set of 2027 *E. faecalis* isolates was performed at the Wellcome Sanger Institute using Illumina-B HiSeq X paired end sequencing platform. Potential mixed strains and species contamination was identified using Kraken v.0.10.6[63]. Assembly and annotation of genome sequences were performed using Velvet assembler and Prokka, respectively, with genus-specific RefSeq database[64–67]. One isolate failed the assembly and annotation pipeline and was, thus, excluded from the analyses run on the assemblies. Mapping was performed using Burrows-Wheeler aligner (BWA-MEM) algorithm[68]. Multi-locus sequence types (MLSTs) were retrieved using SRST2[69], by comparing the Illumina sequence data to the MLST database (https://pubmlst.org/efaecalis)[70].

*Nanopore sequencing and hybrid assemblies.* In order to establish the complete chromosomes and plasmids for a subcollection representative of all the major host types and majority of isolation years, a total of 408 isolates were selected for ONT sequencing. From this set, we early sequenced all *E. faecalis* samples isolated before 1989 (*n* = 42) and performed a hybrid assembly as detailed below. These 42 isolates with a complete genome assembly were considered to label their short-read contigs as plasmid- or chromosome-derived. This allowed to train a machine-learning classifier (support-vector machine) for predicting contigs with a plasmid origin[7]. Using this classifier, we predicted the plasmidome content of all the short-read isolates included in this *E. faecalis* collection. Plasmid predictions were compared using Mash v.2.2.2 distances (*k* = 21, *s* = 1000)[71] and the function hcut (hierarchical clustering, ward.D2 method) from the R package factoextra v.1.0.5[72] indicating 408 clusters (desired number of ONT isolates). Singletons and isolates already long-sequenced were discarded. A random isolate was selected from each cluster, which rendered a total of 242 isolates for ONT sequencing. In addition, we followed the same approach but considering Mash distances (*k* = 21, *s* = 1000) from the total genomic content to select an additional set of 137 *E. faecalis* isolates for ONT sequencing. The ONT runs were completed with additional isolates until reaching 408 isolates. This approach was fundamental to ensure that the subset of isolates selected for long-read sequencing maximised the genomic and plasmid variability present in the entire *E. faecalis* collection.

An initial short-read assembly, using Illumina NextSeq reads, was performed with Unicycler v.0.4.7 (https://github.com/rrwick/Unicycler)[73] with normal mode.

Bioawk v.20110810 (https://github.com/lh3/bioawk) was used to compute the genome size of each isolate considering the initial short-read assembly graph. To reduce the number of ONT reads required to bridge the short-read assembly graph by Unicycler, we undertook a progressive approach considering different subsets of ONT reads equal to a genome coverage starting from 10x to 100x. For this purpose, we used Filtlong (https://github.com/rrwick/Filtlong) which estimated ONT read quality using Illumina NextSeq reads. Filtlong was run indicating: (i) removal of 10% of the worst ONT reads (--keep_percent 90), (ii) the number of long reads to retain from 10x to 100x (--target_bases), (iii) a mean quality weight of 20 (--mean_q_weight) in the ratio between read length/quality and (iv) a minimum ONT read length of 1 kbp.

On the first stage, the subsets of ONT reads were considered to perform a hybrid assembly using Unicycler v.0.4.7. We considered a hybrid assembly as complete if all the components corresponded to circular contigs (circular = TRUE) or corresponded to independent components formed by a single linear contig (circular = FALSE) which could be indicative of a linear extrachromosomal element. Next, if the resulting hybrid assemblies were uncompleted, we added ONT reads mapping to the unbridged paths present in the hybrid assembly graph to perform a consecutive assembly round. The pipeline followed to obtain the hybrid assemblies is publicly available at https://github.com/arredondo23/hybrid_assembly_slurm. Bandage v.0.8.1 (https://github.com/rrwick/Bandage)[74] was used to retrieve the genome assembly statistics corresponding to the number of components, number of dead-ends and N50 present in the completed hybrid assemblies. In total, the hybrid assembly protocol resulted in 335 fully contiguous assemblies.

*Phylogenetic analyses and population structure.* For the species-wide analysis, sequence reads were mapped to the hybrid assembly of one of the oldest isolates in the collection (E07132 "ERX4639547") using BWA[68]. Within the mapping-based core genome, with 2,906,573 bp of reference length, a total of 294,831 SNPs were identified among the 2027 *E. faecalis* isolates by using snp-sites v.2.4.1[75]. The mapping-based SNP alignment was used to infer a ML phylogeny using RaxML v.8.2.8 with the GTR + Gamma rate model[76]. To explore population structure, alignment-free whole-genome clustering through the entire collection was defined by using Population Partitioning Using Nucleotide K-mers (PopPUNK) v.1.2.2[19], using the core-only option. Microreact web application[77] and ggtree2 facet in R v3.6.3 were used to visualise the tree and metadata.

*Pangenome estimation and network analyses.* Pangenomes for the whole collections of 2026 *E. faecalis* and 1602 *E. faecium*[7] isolates that passed the assembly and annotation pipelines and quality control were constructed using Panaroo v.1.2.0 (https://gtonkinhill.github.io/panaroo/#/)[26]. Panaroo was run using its "sensitive" mode with paralog merging enabled. The initial sequence identity threshold was set to 95%, although the Panaroo algorithm allows for clusters below this threshold if they have additional contextual support. Core genes were defined using a 99% presence threshold. The *E. faecalis* pangenome was clustered with Pangenome Neighbour Identification for Bacterial Populations (PANINI) (https://panini.pathogen.watch/)[28], which uses the *t*-distributed stochastic neighbour embedding (*t*-SNE) and Barnes-Hut algorithm. For network analyses of the *E. faecalis* and *E. faecium* accessory genomes (https://github.com/gtonkinhill/Efcm_Efcs_analysis), a graph for each species was generated by using each isolate as a node and connecting two isolates if they shared at least 95% of their accessory genome as defined by Panaroo[26]. Components of the graph that contained less than five isolates were filtered out. Finally, the "organic" layout algorithm in Cytoscape[78] was used to generate a visualisation of the resulting network.

**Evolution of population chromosome and plasmidome.** A second support-vector machine model was trained to predict the origin, chromosome- or plasmid-derived, of *E. faecalis* short-read contigs. We used pentamer frequencies of contigs derived from 298 isolates with a previous complete genome assembly by using mlplasmids (https://gitlab.com/sirarredondo/mlplasmids)[79]. This model achieved a specificity and sensitivity of 0.97 and 0.76, respectively, on an independent set formed by contigs belonging to the remaining long-read completed isolates (*n* = 42). This machine-learning classifier was used to predict the plasmid content of all *E. faecalis* isolates. To evaluate the differences in the genome size either based on hybrid assemblies or on plasmid predictions over the years, the isolates were split in intervals of six (hybrid assemblies) and four (plasmid predictions) consecutive years. The size of the intervals was selected to ensure that each interval consisted of the same number of years. For each interval, we calculated and plotted the mean genome size in bar plots together with all data points present. In this manner, we visualised the distribution of genome sizes over years and summarised the global average genome size differences.

To further investigate the evolutionary changes in the genetic contents of the old *E. faecalis* isolates, plasmid sequences and corresponding chromosome sequences of 30 isolates predating 1985 were separately clustered with Mashtree v.1.0.4[31]. Virulence and resistance genes were annotated with ABRicate v.0.9.8 (https://github.com/tseemann/abricate), with the VFDB database containing 2597 entries[80] and the ResFinder database containing 3077 entries[81], with a minimum coverage of 80% and a minimum identity of 75%. Furthermore, plasmids were

annotated with Prokka v.1.12[66] and the annotation visualised with Clone Manager (Sci Ed Software LLC, Westminster, CO).

**Population history**. Cluster-specific recombination of HA PP clusters (PP2, PP6, PP7, PP18 and PP20) was removed by using Genealogies Unbiased By recomBinations In Nucleotide Sequences (Gubbins) v.2.4.0 (https://github.com/sanger-pathogens/gubbins)[36]. TempEst v.1.5.3[34] with best-fitting root option was used in estimating the quality of temporal signal and least-squares dating (LSD) v.0.3beta[82] and Bayesian Evolutionary Analysis by Sampling Trees (BEAST2) v.2.5.0[83–85] for dating the phylogeny of the HA clusters. Strains with unknown isolation dates were removed from the dating analyses and PP20 due to poor temporal signal as determined by TempEst. BEAST2 analyses were performed using generalised time reversible (GTR) substitution and Gamma site heterogeneity model for 100,000,000 generations, removing 10% burn-in and sampled every 1000 states. Strict clock model was used separately with each of the constant, exponential and Bayesian skyline tree models. Verifying convergence and effective sample size (>200) of three replicate BEAST2 runs per each tree model used and analysing the results were performed in Traces v.1.7.1, and successful replicate runs were combined by using LogCombiner v.2.5.1. Maximum clade credibility trees were inferred in TreeAnnotator v.2.5.1 based on median node height values, and dated trees were visualised with FigTree v.1.4.4.

**Gene content comparison**. In order to compare gene contents between HA and human commensal *E. faecalis* isolates, we performed logistic regression analysis using the mapping-based ML phylogenetic tree to generate a pairwise distance matrix (https://github.com/gtonkinhill/Efcm_Efcs_analysis). Multidimensional scaling (MDS) was used to control for population structure. Genes present in at least 5% and at most 95% of isolates were considered, and *P*-values were adjusted by Bonferroni correction. Of note, overlap in presence between the two categories may exist, while a gene is being called as significant.

**Hospital-associated cluster-specific recombination**. Gubbins v.2.4.0[36] was used to further investigate whether the dated HA PP clusters show any particular signatures of recombination. To identify recombination events putatively undetected by the E07132 alignments, the clusters were additionally aligned to *E. faecalis* V583 reference genome ("AE016830 [https://www.ebi.ac.uk/ena/browser/view/AE016830]"), and elevated sequence diversity regions were identified by Gubbins[36]. Recombination blocks aligned with mapping-based ML phylogeny of the HA clusters were visualised in Phandango (https://jameshadfield.github.io/phandango/#/)[86]. Phages and phage-like elements were identified by Phaster[37] and aligned with the recombination sites on the genome maps of *E. faecalis* E07132 and V583 by using CGView Comparison Tool (https://github.com/paulstothard/cgview_comparison_tool)[87]. Genomic reorganisation within PP clusters was further analysed by using MAUVE[38] in Geneious software package (Biomatters, Ltd., Auckland, New Zealand).

**Antibiotic resistance and virulence gene profiles**. Presence of ARGs and virulence genes was screened directly from sequencing reads with Antimicrobial Resistance Identification By Assembly (ARIBA) v.2.14.4[88], respectively, using ResFinder 3.2 database[81] and VirulenceFinder 2.0 database[89]. ResFinder was supplemented with vancomycin resistance genes *vanA* ("AAA65956.1"), *vanB* ("AAO82021.1"), *vanC* ("AAA24786.1"), *vanD* ("AAD42184.1"), *vanE* ("AAL27442.1"), and *vanG* ("NG_048369.1") and linezolid resistance-conferring *cfrD* ("PHLC01000011"), while virulence gene *hyl*$_{Efm}$ was disregarded in VirulenceFinder as specific to *E. faecium*. The main resultant ARG profiles were collapsed into selected major antibiotic classes: vancomycin (glycopeptide), aminoglycosides, macrolides, lincosamides, tetracyclines, phenicols and oxazolidinones. Both ARG and virulence gene profiles were depicted by aligning with the species-wide reference mapping-based ML phylogeny. To explore resistance to the frontline antimicrobial agent daptomycin, a selection of daptomycin resistance-conferring in-frame deletions in *liaF*, *gdp* and *cls* (respective locus tags EF2913, EF1904 and EF0631 in *E. faecalis* V583 "AE016830.1")[42] were screened by using a custom database in ARIBA.

**Statistical analyses**. To test the hypothesis of the generalist nature of *E. faecalis* across different hosts, empirical CDFs were calculated for relative accessory gene frequencies (0.01–0.99) per each major host type (https://github.com/akpontinen/Efaecalis_eCDF). For comparison within the *E. faecium* population, CDFs were calculated per HA and non-HA host types. Significance between HA and other host types was tested by one-sided permutation tests, with the maximum difference of each two empirical CDFs as a test statistic and 10,000 permutations generated. Differences in means of hybrid assembly chromosome lengths between hospital isolates and other major isolation sources were compared using one-way analysis of variance (ANOVA) test (aov function) and Tukey's honestly significant difference (TukeyHSD function). The analyses were performed in R v.3.6.2 and v.4.0.2[90].

**Reporting summary**. Further information on research design is available in the Nature Research Reporting Summary linked to this article.

## Data availability

Sequence data generated within the study have been deposited at the ENA with accession codes "PRJEB28327" and "PRJEB40976". Descriptive data on the collection of 2027 *E. faecalis* isolates, together with ML phylogeny, PANINI network on Panaroo pangenome output, and temporal and geographic metadata is available within the public Microreact project https://microreact.org/project/3T9X5PlUD. Phylogeny and descriptive data on the old isolate plasmids from 1943 to 1985 are available within the public Microreact project https://microreact.org/project/oR27udmSsi96yeLmL41Wdg. *E. faecium* Illumina reads were retrieved from the ENA "PRJEB28495". All supporting accession codes are available within the article. Source data are provided with this paper, within the custom code repositories detailed in Code availability, and within the Microreact projects. Source data are provided with this paper.

## Code availability

All the custom codes generated within the study are available at public repositories: The codes used to generate the plasmid-related analyses are publicly available at the GitLab repository https://gitlab.com/sirarredondo/efaecalis_plasmids, codes for gene content comparison and accessory genome network analyses are publicly available at the GitHub repository https://github.com/gtonkinhill/Efcm_Efcs_analysis, and codes for empirical CDFs and permutation tests at the GitHub repository https://github.com/akpontinen/Efaecalis_eCDF.

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

## Acknowledgements
A.K.P., R.A.G. and J.C. were funded by Trond Mohn Foundation (Grant TMS2019TMT04), S.A.-A., R.J.L.W. and T.M.C. by the Joint Programming Initiative in Antimicrobial Resistance (JPIAMR2016-AC16/00039), A.R.F. by FCT/MCTES Individual Call to Scientific Employment Stimulus (CEECIND/02268/2017), A.R.F., C.N. and L.P. were funded by the Applied Molecular Biosciences Unit - UCIBIO which is financed by national funds from FCT (UIDP/04378/2020 e UIDB/04378/2020), J.C. also by ERC (Grant 742158) and A.K.P. also by Marie Skłodowska-Curie Actions (Grant 801133).

## Author contributions
C.T., L.P., T.M.C., R.J.L.W. and J.C. designed the study, C.T., L.P., T.M.C. and R.J.L.W. provided the isolates, L.P., A.R.F. and C.N. selected the Portuguese isolates for the study and A.R.F. and C.N. performed their DNA extraction, S.D.B., J.P. and J.C. conducted the sequencing, A.K.P., J.T., S.A.-A., G.T.-H., R.A.G., M.P., R.M., H.P., D.J., J.A.L. and J.C. performed the bioinformatics studies, A.K.P., J.T., S.A.-A., A.C.S., R.J.L.W. and J.C. analysed the data, A.K.P., J.T., R.J.L.W. and J.C. wrote the first draft, all authors contributed to revising the final manuscript.

## Competing interests
The authors declare no competing interests.
