## [Peer Review File · Nature Communications]

REVIEWER COMMENTS

Reviewer #1 (Remarks to the Author):

General:

This is an interesting and generally well-written analysis of the population structure of *E. faecalis* isolated from diverse sources, and a search to discern informative patterns within it. The strength of the manuscript is the highly competent bioinformatic analysis of a large set of *E. faecalis* genomes. There are several points in the manuscript that would benefit from a clearer delineation of terms. First, the natural habitat of *E. faecalis* is the gut environment of a wide range of hosts ranging from invertebrates to humans as established by the work of Mundt and those that followed. It's association with human disease is a rare event. As written, the paper makes the case that *E. faecalis* is a "major human nosocomial pathogen," which is mostly hyperbole – they are mainly commensals that can find their way into the bloodstream, and when they do are difficult to eradicate. This creates something of a strawman later as the implication is that "hospital isolates" represent a group associated with disease pathogenesis. This reviewer did not find a case being made that these are isolates from sites of human infection, and strongly suspects that the majority may simply be human commensal isolates from surveillance cultures. If that is indeed the case, a pathogen-associated signal in a dataset of this size could easily be diluted to a point where it would not fall below a stringent statistical threshold. "Hospital strain" collection bias is also a problem in comparing population structures of *E. faecalis* and *E. faecium*, which is a limitation of the interpretations made in the manuscript. Specifically, *E. faecium* as the paper indirectly notes, is much more likely to be vancomycin resistant. It is also less common as a commensal microbe in humans. As a result, it is more likely to be isolated in hospitals because of treatment failure, and hospital isolates of *E. faecium* are more likely to exhibit a population structure driven by the presence of resistance-bearing mobile elements. There certainly is more to it than that, but great care needs to be taken in making fairly blind comparisons. The bottom line is that the most problematic and pathogenic lineages of enterococci are those having acquired resistances and factors that enhance colonization or the propensity to cause disease that are not core to the species, lineages that circulate mainly within hospitals and are rare in the community, not those isolated in hospital from surveillance cultures which are likely enriched and potentially predominated by commensal strains, or even singletons that cause rare outpatient infections such as endocarditis or UTI that ultimately may be treated in hospital. It is difficult to see this important distinction being made in the analysis within, and this is a point that should be clarified.

A second point that seems ambiguous if not incorrect, is the interpretation of the study that examined the origins of the genus *Enterococcus*. Lines 93 – 96 seem to suggest it was a study of MDR hospital isolates of *E. faecalis* and *E. faecium*. This reviewer's understanding is that a wide representation of enterococcal species were studied, including commensal and clinical representatives of *E. faecalis* and *E. faecium*, and the survival traits that positioned select lineages of those species to emerge as potential pathogens capable of endemic residence and spread within hospitals were common to all or nearly all enterococci. This becomes important in understanding the later reference to the origin of HA clusters (lines 377 – 393). The prediction from that previous report would be that the traits that favor survival in harsh environments present in all or nearly all enterococcal species, would not be those that would lead to the genesis of new sequence types associated with hospital endemnicity, but would be necessary precursors very likely also associated with their generalist success in many hosts as observed decades ago by Mundt and others. That is, sequence types related to those found now in hospitals should be represented in older collections, and might even be enriched among lineages to which humans are most frequently exposed to. The data presented doesn't seem to change that view. Some global assessments of gene content between the common hospital clusters observed here and elsewhere are made, but a rigorous comparison of specific gene content between these lineages and commensal isolates from the community is not clearly laid out.

The inclusion of isolates from wild birds is a nice augmentation of the collection. However, it is common to observe flocks of seagulls feeding in garbage dumps, or Canada geese feeding in city parks, so just as it is important to discern infection-associated isolates from commensal surveillance isolates for hospital associated strains, for comparing human-associated *E. faecalis* with wild animal-associated *E. faecalis*, it is important to distinguish birds that spend substantial feeding time in ecosystems dominated by humans from those that feed at sites free from human influence. The authors refer to findings from a recent study of marine penguins that inhabit an ecology with extremely little human influence. It is unlikely that that level of isolation also applies to the birds from which isolates were obtained here, but that should be clarified. The point is that "non-human origin" or "wild" for isolate sources doesn't equate with lack of human influence. In fact the authors note that the paper on penguin isolates referred to failed to find anything but intrinsic AMR genes, but then make the general statement that appears to be an overinterpretation that "...antibiotic usage and concomitant selective pressure specifically in wild birds is indeed negligible...". In this reviewer's view, given global levels of pharmaceutical pollution, that statement currently likely only applies to birds in the Antarctic and few other extremely isolated ecosystems, and specifically not to birds that even temporarily share human-influenced habitats – from city dumps, to crop fields, to orchards, vineyards and other agricultural sites. The extent to which the presence of non-core genome AMR genes identified in this wild bird set is influenced by human activity, or somehow reflects a level of naturally circulating resistance in truly wild ecologies seems highly arguable without clearer information on the provenance of those isolates and the habits of their hosts.

The bottom line is that 1) compared to its abundance in nature, and specifically its abundance as a commensal in humans, *E. faecalis* is a rare cause of disease, 2) all hospital isolates are not necessarily infection-derived or hospital endemic, and 3) a bird may be wild, but its enterococci may still be heavily influenced by human activity including widespread use of chemically diverse agents with antimicrobial activity.

All of that said, these points do not detract from the technical sophistication and rigor of the analyses conducted, only their interpretation. The extensive analyses made add substantially by filling important knowledge gaps in the field.

Specific:

The manuscript is well edited and the data are well presented in this reviewer's view.

Reviewer #2 (Remarks to the Author):

The study by Pontinen AK. et al. shows the analyses of 2,027 genomes of *E. faecalis* isolates collected over a time span of 82 years and recovered from human and non-human sources in different countries. The authors provide robust evolutive analyses evidencing that *E. faecalis* behaves as a generalist organism. This is a novel and interesting study that has not been previously performed. Moreover, the contribution of genomic data of such as human pathogen is valuable. Even though, the methods are sound, authors could provide deeper analysis and some conclusions should be considered according the offered results. Additionally to the genomic analysis and the presented results, authors could perform an analysis of the virulence and antibiotic resistance traits potentially contributing to the nosocomial adaptation of *E. faecalis*.

I have few suggestions to improve the clarity of the manuscript:

Line 64: the assertion of a stable core genome size is not supported by the linear regression analysis, as the model is not adjusted to the data (very low R squared values) and high variance among samples. Please rephrase this to meet the data.

Line 138: Since there is a correlation between ST and PPs, authors could consider to represent this correlation in the figure 1.

Line 147: The use of the core genome in a population of genomes so diverse and extensive would limit the available ortholog groups to identify host specificity among the groups, thus reducing the probability of identifying deep branches in the phylogeny, in particular when using a high threshold (99%) to include an ortholog group in the analysis. Please acknowledge this possible limitation in the analysis and the result.

Line 159: Please show Heap's Law alpha value to identify if the pangenome is an open or closed one.

Lines 167-168: By definition, PopPUNK is kmer based analysis not restricted to core genome, please clarify and correct the sentence accordingly.

Lines 189-209: As mentioned before, the results presented in the whole section are of limited value, as the fit of the linear regression of the genome size through time is very poor (very low R squared values). Additionally, the plot clearly does not show a trend among the samples but high variance. Please reevaluate the model, a transformation of the genome size into log scale or kb scale, rather than pb may help. Also, comparison of regression models of the genome sizes across hosts would strengthen the analysis if the data shows that there is no difference among them.

Lines 210-252: It is interesting the high diversity of plasmids found in the collection, even from early isolates. How those plasmids expanded or disappeared in the more recent isolates? Where clusters correlated to the presence/absence of such plasmids?

Line 252: is the word "clones" correct? or should it be "strains" or "lineages"?

Lines 280-308: Were the recombinant regions composed only by phages sequences? Were any genes between those flanking regions and if they were, what possible functions were present? Would those be able to give any ecological advantage/disadvantage?

Lines 360-362: That is what would be expected from the analysis of the genomic data, but no functional association among the collection was performed. Is this true for this data? Are wide molecular functions present in all the identified clusters? Are those functions host-associated or not? The presence of mercury and arsenic resistance genes in specific PP would show a little specialization among those groups (lines 243-252). Is there a way to measure how generalist or specialist is a genome?

RESPONSE TO REVIEWERS

We kindly thank the reviewers for their valuable comments and insightful review of our manuscript entitled “Apparent nosocomial adaptation of the generalist major pathogen *Enterococcus faecalis* predates the modern hospital era”. We have revised our manuscript accordingly with tracked changes in the enclosed marked-up manuscript. Please find point-by-point answers to the remarks made by the reviewers below. We hope that these will satisfyingly address the justified issues arisen within our manuscript. We also wish to point out that, in order to avoid drawing unwarranted conclusions, we have accordingly modified the title of our manuscript: “Apparent nosocomial adaptation of *Enterococcus faecalis* predates the modern hospital era”. As additional amendments, we have included hybrid assembly locations in ‘Data availability’ and added a separate ‘Code availability’ section.

Reviewer #1 (Remarks to the Author):

General:

This is an interesting and generally well-written analysis of the population structure of *E. faecalis* isolated from diverse sources, and a search to discern informative patterns within it. The strength of the manuscript is the highly competent bioinformatic analysis of a large set of *E. faecalis* genomes. There are several points in the manuscript that would benefit from a clearer delineation of terms. First, the natural habitat of *E. faecalis* is the gut environment of a wide range of hosts ranging from invertebrates to humans as established by the work of Mundt and those that followed. It’s association with human disease is a rare event. As written, the paper makes the case that *E. faecalis* is a “major human nosocomial pathogen,” which is mostly hyperbole – they are mainly commensals that can find their way into the bloodstream, and when they do are difficult to eradicate. This creates something of a strawman later as the implication is that “hospital isolates” represent a group associated with disease pathogenesis. This reviewer did not find a case being made that these are isolates from sites of human infection, and strongly suspects that the majority may simply be human commensal isolates from surveillance cultures. If that is indeed the case, a pathogen-associated signal in a dataset of this size could easily be diluted to a point where it would not fall below a stringent statistical threshold. “Hospital strain” collection bias is also a problem in comparing population structures of *E. faecalis* and *E. faecium*, which is a limitation of the interpretations made in the manuscript. Specifically, *E. faecium* as the paper indirectly notes, is much more likely to be vancomycin resistant. It is also less common as a commensal microbe in humans. As a result, it is more likely to be isolated in hospitals because of treatment failure, and hospital isolates of *E. faecium* are more likely to exhibit a population structure driven by the presence of resistance-bearing mobile elements. There certainly is more to it than that, but great care needs to be taken in making fairly blind comparisons. The bottom line is that the most problematic and pathogenic lineages of enterococci are those having acquired resistances and factors that enhance colonization or the propensity to cause disease that are not core to the species, lineages that circulate mainly within hospitals and are rare in the community, not those isolated in hospital from surveillance cultures which are likely enriched and potentially predominated by commensal strains, or even singletons that cause rare outpatient infections such as endocarditis or UTI that ultimately may be treated in hospital.

It is difficult to see this important distinction being made in the analysis within, and this is a point that should be clarified.

We thank the reviewer for positive remarks concerning our study and agree that emphasizing ‘major human pathogen’ in the title was unwarranted and consequently we have edited the title accordingly. However, there has been a misunderstanding concerning the primary source of hospital samples and we now more clearly point out the fact that the majority of hospital isolates in our collection are indeed blood culture isolates causing clinical manifestations and thereby represent true infections rather than random carriage, which lends support to the conclusions made in the study. Please see Lines 116-119 in the revised manuscript for more detailed sample description of human hospital isolates. We also wish to point out that we have now referred to the pioneering work by Mundt and others (Mundt, 1963, *Appl Microbiol* and Martin and Mundt, 1972, *Appl Microbiol*), describing the wide distributions of enterococci in animals and insects (Line 94 in the revised manuscript). Furthermore, we have now more directly indicated the prevalence of vancomycin resistance in *E. faecium* as opposed to *E. faecalis* (Lines 322-324 in the revised manuscript).

A second point that seems ambiguous if not incorrect, is the interpretation of the study that examined the origins of the genus *Enterococcus*. Lines 93 – 96 seem to suggest it was a study of MDR hospital isolates of *E. faecalis* and *E. faecium*. This reviewer’s understanding is that a wide representation of enterococcal species were studied, including commensal and clinical representatives of *E. faecalis* and *E. faecium*, and the survival traits that positioned select lineages of those species to emerge as potential pathogens capable of endemic residence and spread within hospitals were common to all or nearly all enterococci.

We thank the reviewer for pointing out this ambiguity, our intention was not to give such an impression when citing the important study by Lebreton *et al.* and have now reworded the text in the revision to avoid this.

This becomes important in understanding the later reference to the origin of HA clusters (lines 377 – 393). The prediction from that previous report would be that the traits that favor survival in harsh environments present in all or nearly all enterococcal species, would not be those that would lead to the genesis of new sequence types associated with hospital endemicity, but would be necessary precursors very likely also associated with their generalist success in many hosts as observed decades ago by Mundt and others. That is, sequence types related to those found now in hospitals should be represented in older collections, and might even be enriched among lineages to which humans are most frequently exposed to. The data presented doesn’t seem to change that view. Some global assessments of gene content between the common hospital clusters observed here and elsewhere are made, but a rigorous comparison of specific gene content between these lineages and commensal isolates from the community is not clearly laid out.

To address the remark made by the reviewer, we performed the suggested gene content comparison of hospital-associated and commensal *E. faecalis* isolates. Consequently, we found multiple genes of interest enriched in the hospital isolates, although none of the

genes were completely absent in the commensal ones. We have further elaborated this in the results chapter 'Early emergence of hospital-associated lineages'.

The inclusion of isolates from wild birds is a nice augmentation of the collection. However, it is common to observe flocks of seagulls feeding in garbage dumps, or Canada geese feeding in city parks, so just as it is important to discern infection-associated isolates from commensal surveillance isolates for hospital associated strains, for comparing human-associated *E. faecalis* with wild animal-associated *E. faecalis*, it is important to distinguish birds that spend substantial feeding time in ecosystems dominated by humans from those that feed at sites free from human influence. The authors refer to findings from a recent study of marine penguins that inhabit an ecology with extremely little human influence. It is unlikely that that level of isolation also applies to the birds from which isolates were obtained here, but that should be clarified. The point is that "non-human origin" or "wild" for isolate sources doesn't equate with lack of human influence. In fact the authors note that the paper on penguin isolates referred to failed to find anything but intrinsic AMR genes, but then make the general statement that appears to be an overinterpretation that "...antibiotic usage and concomitant selective pressure specifically in wild birds is indeed negligible...". In this reviewer's view, given global levels of pharmaceutical pollution, that statement currently likely only applies to birds in the Antarctic and few other extremely isolated ecosystems, and specifically not to birds that even temporarily share human-influenced habitats – from city dumps, to crop fields, to orchards, vineyards and other agricultural sites. The extent to which the presence of non-core genome AMR genes identified in this wild bird set is influenced by human activity, or somehow reflects a level of naturally circulating resistance in truly wild ecologies seems highly arguable without clearer information on the provenance of those isolates and the habits of their hosts.

We thank the reviewer for pointing out the need to clarify the inclusion of wild bird isolates in the study and the origin and species of the birds. The wild bird collection represented 11 different avian orders, and the vast majority of the isolates was retrieved from orders and species inhabiting ecosystems at a distance from the immediate influence of human domination. Many of the species are also known to be migratory. Only a minority of isolates was collected from species, such as gulls and mallard, that commonly inhabit and feed in areas within or in close contact with human settlements. We have clarified these aspects in the description of the isolate collection, please see Lines 122-127 in the revised manuscript. In addition, we fully agree with the notion that even wild birds may be widely influenced by human activity through shared habitats, and it was indeed not our intention to make such a claim that this would not be the case. We have thereby now removed the ambiguous statement and more carefully formulated the discussion.

The bottom line is that 1) compared to its abundance in nature, and specifically its abundance as a commensal in humans, *E. faecalis* is a rare cause of disease, 2) all hospital isolates are not necessarily infection-derived or hospital endemic, and 3) a bird may be wild, but its enterococci may still be heavily influenced by human activity including widespread use of chemically diverse agents with antimicrobial activity.

All of that said, these points do not detract from the technical sophistication and rigor of the analyses conducted, only their interpretation. The extensive analyses made add substantially by filling important knowledge gaps in the field.

Specific:

The manuscript is well edited and the data are well presented in this reviewer's view.

Reviewer #2 (Remarks to the Author):

The study by Pontinen AK. et al. shows the analyses of 2,027 genomes of *E. faecalis* isolates collected over a time span of 82 years and recovered from human and non-human sources in different countries. The authors provide robust evolutive analyses evidencing that *E. faecalis* behaves as a generalist organism. This is a novel and interesting study that has not been previously performed. Moreover, the contribution of genomic data of such as human pathogen is valuable.

Even though, the methods are sound, authors could provide deeper analysis and some conclusions should be considered according the offered results. Additionally to the genomic analysis and the presented results, authors could perform an analysis of the virulence and antibiotic resistance traits potentially contributing to the nosocomial adaptation of *E. faecalis*.

We thank the reviewer for positive feedback concerning our study. As suggested, we have now performed additional virulence gene analysis. On top of the resistance traits, the virulence traits were also compared between the nosocomial and other isolates to investigate whether the two showed systematic differences. Methods and results were revised accordingly in chapters 'Antibiotic resistance and virulence gene profiles' and 'Early emergence and dissemination of antimicrobial resistance and virulence factors in the *E. faecalis* population', respectively, and the virulence gene screening results were also added into Supplementary Table 3 and presented as Supplementary Figure 16.

I have few suggestions to improve the clarity of the manuscript:

Line 64: the assertion of a stable core genome size is not supported by the linear regression analysis, as the model is not adjusted to the data (very low R squared values) and high variance among samples. Please rephrase this to meet the data.

We have revised the adopted methods here to more profoundly show that the core genome sizes indeed remain stable over time and different sources. Please see revised Figure 4a and Supplementary Figure 6 as well as revised methods in chapter 'Evolution of population chromosome and plasmidome'.

Line 138: Since there is a correlation between ST and PPs, authors could consider to represent this correlation in the figure 1.

We have revised Figure 1 as suggested and have accordingly implemented the major sequence types as a separate panel in the figure.

Line 147: The use of the core genome in a population of genomes so diverse and extensive would limit the available ortholog groups to identify host specificity among the groups, thus reducing the probability of identifying deep branches in the phylogeny, in particular when using a high threshold (99%) to include an ortholog group in the analysis. Please acknowledge this possible limitation in the analysis and the result.

We kindly apologize for the misunderstanding arisen within the description of phylogeny in the current context, however the phylogeny was indeed mapping-based core genome construction, opted for in order to avoid such potential limitations justly pointed out by the reviewer. We sought to accordingly clarify this in the results and methods.

Line 159: Please show Heap's Law alpha value to identify if the pangenome is an open or closed one.

Heap's Law alpha value of 2.16, suggesting a closed pangenome for *E. faecalis*, was added to the results.

Lines 167-168: By definition, PopPUNK is kmer based analysis not restricted to core genome, please clarify and correct the sentence accordingly.

PopPUNK is indeed not restricted to core genome, and outputs both core and accessory distances. Clusters in the present study were, however, defined using only the core distances (--core-only option), and additional accessory distance clustering was performed by using PANINI. We sought to clarify this aspect in the results, please see Lines 168-169 in the revised manuscript.

Lines 189-209: As mentioned before, the results presented in the whole section are of limited value, as the fit of the linear regression of the genome size through time is very poor (very low R squared values). Additionally, the plot clearly does not show a trend among the samples but high variance. Please reevaluate the model, a transformation of the genome size into log scale or kb scale, rather than pb may help. Also, comparison of regression models of the genome sizes across hosts would strengthen the analysis if the data shows that there is no difference among them.

We have removed the results based on the linear model and replaced them with distribution plots on temporally binned data to demonstrate the lack of any trend and persistent variability as pointed out by the reviewer, please see the revised Figure 4a and Supplementary Figure 6.

Lines 210-252: It is interesting the high diversity of plasmids found in the collection, even from early isolates. How those plasmids expanded or disappeared in the more recent isolates? Where clusters correlated to the presence/absence of such plasmids?

We agree that this would indeed be an intriguing question to explore further. However, in order to expand the study to cover the more recent isolates and to in-depth elaborate the evolutionary trends of plasmids over the recent years, we would unfortunately need to quite significantly breach the given word limits within the current manuscript. We do feel

that the subject would be worth a more thorough follow-up in itself and have accordingly warranted further analyses for a more comprehensive and detailed picture.

Line 252: is the word “clones” correct? or should it be “strains” or “lineages”?

We agree with the notation and have reworded “clones” as “lineages”.

Lines 280-308: Were the recombinant regions composed only by phages sequences? Were any genes between those flanking regions and if they were, what possible functions were present? Would those be able to give any ecological advantage/disadvantage?

As indicated in the text, besides phage sequences we identified (putative) integrated plasmids at the recombination site. We unfortunately found no specific genes of similar functions in the flanking regions that might have explained any ecological advantages. This dissimilarity might be expected, as integration of phages occurs via attachment sites and does not require any other specific genes in the vicinity of the integration.

Lines 360-362: That is what would be expected from the analysis of the genomic data, but no functional association among the collection was performed. Is this true for this data? Are wide molecular functions present in all the identified clusters? Are those functions host-associated or not? The presence of mercury and arsenic resistance genes in specific PP would show a little specialization among those groups (lines 243-252). Is there a way to measure how generalist or specialist is a genome?

We performed an additional gene content comparison of hospital-associated and commensal *E. faecalis* isolates in order to more thoroughly investigate potential functions associated with hospital settings as opposed to commensal environment. We found a number of genes of interest, particularly within mobile genetic elements and more specifically linked to *E. faecalis* pathogenicity island, significantly enriched in the hospital isolates. However, none of these genes were strictly present in the hospital isolates while completely absent in the others. We see this suggesting that these enriched genes themselves are not strictly required for survival in hospital settings. While previous studies, e.g. meta-analysis of microbial communities across various environments (Sriswasdi *et al.*, 2017, *Nat Comm*) have shown significantly larger genomes for generalists, yet in a specific individual case, a decisive limit differentiating between a generalist and a specialist might be difficult to assess.

REVIEWERS' COMMENTS

Reviewer #1 (Remarks to the Author):

The authors have addressed the issues previously raised and have revised the manuscript accordingly. This reviewer has no further suggestions for improvement.